# Determination of the Elution Capacity of Dalbavancin in Bone Cements: New Alternative for the Treatment of Biofilm-Related Peri-Prosthetic Joint Infections Based on an *In Vitro* Study

**DOI:** 10.3390/antibiotics11101300

**Published:** 2022-09-23

**Authors:** Mar Sánchez-Somolinos, Marta Díaz-Navarro, Antonio Benjumea, Marta Tormo, José Matas, Javier Vaquero, Patricia Muñoz, Pablo Sanz-Ruíz, María Guembe

**Affiliations:** 1Department of Clinical Microbiology and Infectious Diseases, Hospital General Universitario Gregorio Marañón, 28007 Madrid, Spain; 2Instituto de Investigación Sanitaria Gregorio Marañón, 28007 Madrid, Spain; 3Department of Orthopaedic Surgery and Traumatology, Hospital General Universitario Gregorio Marañón, 28007 Madrid, Spain; 4School of Biology, Universidad Complutense de Madrid, 28040 Madrid, Spain; 5School of Medicine, Universidad Complutense de Madrid, 28040 Madrid, Spain; 6CIBER Enfermedades Respiratorias-CIBERES (CB06/06/0058), 28029 Madrid, Spain

**Keywords:** dalbavancin, bone cement, polymethylmethacrylate, elution, cumulative concentration, vancomycin

## Abstract

Antibiotic-loaded bone cement is the most widely used approach for the treatment of biofilm-induced septic sequelae in orthopedic surgery. Dalbavancin is a lipoglycopeptide that acts against Gram-positive bacteria and has a long half-life, so we aimed to assess whether it could be a new alternative drug in antibiotic-loaded bone cement for the treatment of periprosthetic joint infections. We assessed the elution capacity of dalbavancin and compared it with that of vancomycin in bone cement. Palacos^®^R (Heraeus Medical GmbH, Wehrheim, Germany) bone cement was manually mixed with each of the antibiotics studied at 2.5% and 5%. Three cylinders were obtained from each of the mixtures; these were weighed and incubated in 5 mL phosphate-buffered saline at 37°C under shaking for 1 h, 2 h, 4 h, 8 h, 24 h, 48 h, 168 h, and 336 h. PBS was replenished at each time point. The samples were analyzed using high-performance liquid chromatography (vancomycin) and mass cytometry (dalbavancin). Elution was higher than the minimum inhibitory concentration (MIC)90 for both antibiotics after 14 days of study. The release of vancomycin at 14 days was higher than of dalbavancin at each concentration tested (*p* = 0.05, both). However, the cumulative release of 5% dalbavancin was similar to that of 2.5% vancomycin (*p* = 0.513). The elution capacity of dalbavancin reached a cumulative concentration similar to that of vancomycin. Moreover, considering that the MIC90 of dalbavancin is one third that of vancomycin (0.06 mg/L and 2 mg/L, respectively) and given the long half-life of dalbavancin, it may be a new alternative for the treatment of biofilm-related periprosthetic infections when loaded in bone cement.

## 1. Introduction

The most notable complication in orthopedic surgery is infection. Among these, the most important are peri-prosthetic joint infection (PJI), septic pseudoarthrosis (absence of fracture healing due to infection), and chronic osteomyelitis (with an unknown incidence due to its late presentation) [1,2,3]. In particular, the treatment of PJI remains the same as it was 20 years ago, with a combination of a surgical approach and systemic and local antibiotic treatment [3,4]. The most common causative agents of PJI are Gram-positive bacteria capable of forming a biofilm that makes the infection difficult to eradicate [1].

Regarding the use of local antibiotics, many strategies have been studied to achieve optimal release at the site of infection, with bone cement (polymethylmethacrylate [PMMA]) being the most widely used tool due to the extensive experience in its use, as well as its high safety profile. Furthermore, its use in the Masquelet technique (in the first stage, creation of vascularized membranes in growth factors with the implantation of a bone cement with antibiotics that resolves the infection, and in the second stage, implantation in these membranes of a bone graft that favors the reconstruction of the bone defect) is crucial not only for its usefulness as a vehicle for antibiotics but also for its capacity to generate induced membranes, necessary for the second stage of this technique. However, some of the antibiotics conventionally used have unfavorable characteristics for local use in PMMA (low release, short half-life, etc.) [5,6,7,8,9,10,11,12]. Even so, given the absence of more effective alternatives, their use remains the gold standard for the treatment of bone infections. 

Nau et al. showed that the thickness and quality of the induced membranes depend on the cement and antibiotic used [7]. Several authors have observed that the use of other vehicles for the development of these membranes, such as calcium sulphates, were not effective for the correct development of this technique, with the use of bone cement (PMMA) being necessary to obtain a good result [13,14,15,16].

Dalbavancin is an antibiotic belonging to the group of semi-synthetic lipo-glycopeptides and is structurally related to teicoplanin. Like other glycopeptides, the mechanism of action of dalbavancin is interference with bacterial cell wall formation by binding to the D-alanine-D-alanine terminal end of the peptidoglycan, preventing its elongation [17,18]. It is active against Gram-positive, aerobic, and anaerobic bacteria and has shown in pharmacodynamic studies that, compared to other structurally related antibiotics such as vancomycin and teicoplanin, it has greater activity against methicillin-susceptible *Staphylococcus aureus* (MSSA), methicillin-resistant *S. aureus* (MRSA), vancomycin-intermediate susceptible *S. aureus* (VISA), and *Staphylococcus epidermidis* [18]. Regarding its activity against Staphylococci biofilms, it has also been demonstrated to be better than vancomycin [19,20,21,22].

Due to its pharmacokinetic characteristics, and in particular its long half-life (372 h), dalbavancin can be administered at least once a week following a regimen of one 1000 mg and one 500 mg dose spaced 7 days apart [23]. Some authors have even demonstrated that its efficacy can last up to 14 days [24]. Although currently approved for the treatment of skin and soft tissue infections, the application in the treatment of PJI is under extensive investigation [25,26,27,28,29,30,31,32]. Moreover, it has been demonstrated that it is cost-effective [33,34,35]. Therefore, the use of dalbavancin loaded in bone cement may be a new alternative that needs to be further investigated.

Therefore, we aimed to assess the elution capacity of dalbavancin at 2.5% and 5% and compare it to vancomycin at the same concentrations during a 14-day period.

## 2. Results

The cumulative antibiotic release over the 14-day period for the four groups is detailed in Figure 1. The mean ± standard deviation (SD) of each antibiotic at each concentration during the 14-day study period are detailed in Table 1. Vancomycin and dalbavancin eluted after 14 days containing the same amount of antibiotic (2.5% and 5%) showed significant differences, being greater for vancomycin (*p* = 0.05 in both cases). However, when 5% dalbavancin was compared to 2.5% vancomycin, it showed a similar release at the end of the study period (cumulative concentration mean ±SD: 157.12 ± 12.43 and 158.49 ± 3.09, respectively; *p* = 0.513).

In summary, the elution from both antibiotics was higher than the minimum inhibitory concentration (MIC)90 during the entire study period.

## 3. Discussion

Our research is the first in vitro study describing the elution capacity of dalbavancin in bone cements, which showed similar kinetics to vancomycin.

PJI is one of the most relevant complications in orthopedic surgery [3], which is well known to be a biofilm-related infection [36,37]. The use of local antibiotics is one of the recommended approaches to treat PJI, antibiotic-loaded bone cements being the most widely used [8,38,39,40,41] and, in particular, dual-antibiotic high-loaded bone cements are more effective and cost efficient than single ones for the prevention of PJI [33]. One of the most used antibiotics is vancomycin, which has shown a suitable elution capacity and antimicrobial activity in several studies [42,43,44]. Hsu et al. showed that vancomycin- and ceftazidime-loaded bone cements provided broad-spectrum antibacterial capacity both in vitro and in vivo [45]. Paz et al. had a cumulative concentration of 150 µg/mL with 10% vancomycin at 336 h [8], which correlates similarly with our findings, as our cumulative concentration of 5% vancomycin was 192.67 µg/mL at the same time.

However, several limitations have been described. One important limitation is that the anti-biofilm activity of the currently used antibiotics loaded in bone cements is insufficient [46,47]. Therefore, we aimed to search for new alternative antibiotics, such as dalbavancin, which has demonstrated to have high biofilm activity [19,20,25,28,30,48,49]. Despite only being recommended for the treatment of skin and soft tissue infections, it revealed promising results for its use in bone and joint infections [22,26,28,29,30,31,50,51,52]. Another limitation of antibiotic-loaded bone cements is that mechanical characteristics of cements may be impaired due to high levels of antibiotic eluted in the cement or the use of combined antibiotics. Levack et al. identified notable differences in thermal stability and elution among tobramycin, amikacin, fosfomycin, minocycline, and meropenem [53]. Slane et al. suggested that tobramycin eluted more effectively than vancomycin and that high antibiotic loading in cement does not necessarily lead to enhanced antibiotic elution [54]. Klekamp et al. showed that the fatigue life of bone cement was significantly decreased with vancomycin alone and in combination with tobramycin [55]. Paz et al. demonstrated that cements with cefazolin showed much higher elution than those containing the same concentration of vancomycin, whereas the strength was lower. This may be explained because the size of the molecule will directly affect its diffusion; as cefazolin has a lower molecular weight than vancomycin, small crystals of cefazolin may increase the fluid uptake and promote its diffusion [8]. This correlates with our findings, as dalbavancin, which has a higher molecular weight than vancomycin, showed slightly lower kinetic elution at 2.5%. Another limitation is the short time that some antibiotics remain above the MIC, a fact that is overcome by dalbavancin, thanks to its high half-life. In addition, systemic toxicity and the promotion of antibiotic resistance in prolonged spacer retention are also other potential complications. 

Based on our results, dalbavancin showed a 46% and 20% lower elution rate than vancomycin at the same doses added to the PMMA (2.5% and 5%, respectively). However, as dalbavancin MIC90 is 16.6-times lower than vancomycin, its half-life is significantly higher than vancomycin (372 h), and it has greater efficacy against biofilm, we suggest that dalbavancin can be an alternative for loaded bone cements [18,25,56]. Despite this, our study has the limitation that we only tested a 14-day period. Therefore, future studies are needed to test dalbavancin efficacy for longer periods and against Staphylococcal biofilms using in vitro models of bone cement elution. In addition, it is also important to further investigate the impact of dalbavancin-loaded cement on the mechanical properties of cement and potential toxicity.

## 4. Materials and Methods

The study was carried out in a tertiary teaching hospital in Madrid, Spain.

### 4.1. Preparation of the Antibiotic-Loaded Bone Cements

Palacos^®^ R bone cements (Heraeus Medical GmbH, Wehrheim, Germany) were manually mixed with each of the antibiotics studied (vancomycin and dalbavancin), from the vial powder prepared for routine intravenous use, in proportions of 2.5% (1 g of antibiotic per 40 g of cement) and 5% (2 g of antibiotic per 40 g of cement). Three cylinders of 1 cm diameter and 0.5 cm were prepared from each of the mixtures using 5 mL syringes by hand (Appendix A). The weight of each cylinder was determined and incubated in 5 mL phosphate-buffered saline (PBS) at 37 °C under shaking (150 rpm) for 1 h, 2 h, 4 h, 8 h, 24 h, 48 h, 168 h, and 336 h. At each time, 2 mL aliquots were obtained and the cylinders were re-incubated in another tube with 5 mL fresh PBS. Aliquots from each time were frozen for further study by high-performance liquid chromatography (vancomycin) and mass cytometry (dalbavancin) (Appendix A).

### 4.2. Vancomycin Analysis

Samples from release studies of vancomycin were analyzed as received using an HPLC with a Varian Prostar 230 Solvent Delivery Module, Varian Prostar autosampler 410, and Varian Prostar 310 UV–visible detector (Varian, Palo Alto, CA, USA). Peak integration was analysed using the Galaxie software (version 1.9). Vancomycin was analyzed using a Nucleosil C18 column (250 mm × 4.6 mm, 5 μm). The mobile phase consisted of 50 mM ammonium phosphate: acetonitrile (92:8, *v*:*v*), with a final pH of 2.2, which was pumped at a flow rate of 0.7 mL/min and the injection volume of the sample was 40 µL. The column temperature was maintained at 25 °C and the detector was set at 205 nm.

### 4.3. Dalbavancin Analysis

Samples from release studies of dalbavancin were analyzed as received using LC-ESI-QQQ-MS (LC-8030 Shimadzu, Manchester, U.K.). Dalbavancin was analyzed using a Phenomenex Gemini C18 columns (110 A 150 mm × 2 mm, 5 μm). The gradient mode consisted of 5% Phase B—7 min 95% Phase B—8 min 95% Phase B—8.5 min 5% Phase B. Phase A consisted of H_2_O + 0.1% formic acid and Phase B of CAN. It was pumped at a flow rate of 0.4 mL/min and the injection volume of the sample was 40 µL. The run time was 10 min and the MRM transitions for D were: Quantifier (*m*/*z*): 909.1 > 340.0 CE: −36; Qualifier (*m*/*z*): 909.1 > 730.5 CE: −26.

The cumulative release of both antibiotics over the 14 days was calculated. The elution testing was performed in triplicate.

### 4.4. Statistical Analysis

For the comparison of the cumulative concentration mean between vancomycin and dalbavancin at the two doses, we used Student’s *t*-test.

Statistical significance was set at *p* < 0.05 for all the tests. The statistical analysis was performed using IBM SPSS Statistics for Windows, Version 21.0 (IBM Corp, Armonk, NY, USA).

## 5. Conclusions

To our knowledge, this is the first in vitro study describing the potential use of dalbavancin-loaded bone cement to be considered in the management of PJI, as similar elution cumulative concentration was observed to that of vancomycin.

## Figures and Tables

**Figure 1 antibiotics-11-01300-f001:**
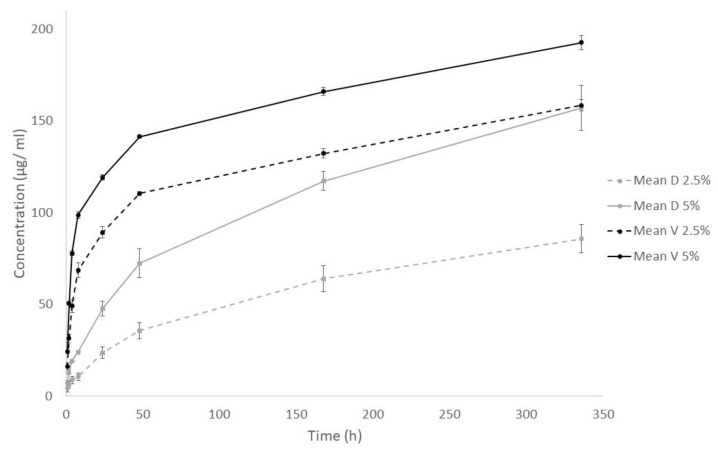
Cumulative antibiotic release over 14-day period for each group of antibiotics. V, vancomycin; D, dalbavancin; h, hours.

**Table 1 antibiotics-11-01300-t001:** Cumulative concentration for each antibiotic over a 14-day period.

Time (Hours)	Cumulative Concentration (µg/mL) Mean ± SD			
	2.5% V	5% V	2.5% D	5% D
1	16.13 (1.88)	24.44 (0.42)	4.35 (1.92)	7.43 (1.18)
2	31.49 (2.44)	50.54 (0.77)	6.32 (2.00)	13.05 (1.30)
4	49.08 (3.56)	77.95 (1.37)	8.75 (1.97)	19.25 (0.39)
8	68.60 (4.15)	98.68 (0.86)	10.58 (1.91)	23.98 (0.56)
24	89.31 (3.25)	119.27 (1.39)	23.70 (3.13)	47.82 (3.96)
48	110.55 (1.07)	141.48 (0.57)	35.68 (4.35)	72.45 (7.71)
168	132.29 (2.65)	166.07 (2.28)	63.98 (7.08)	117.32 (5.23)
336	158.49 (3.09)	192.67 (3.92)	85.73 (7.64)	157.12 (12.43)

SD, standard deviation, V, vancomycin; D, dalbavancin.

## Data Availability

Not applicable.

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
