# Peer review of "Determination of the Elution Capacity of Dalbavancin in Bone Cements: New Alternative for the Treatment of Biofilm-Related Peri-Prosthetic Joint Infections Based on an In Vitro Study"

_antibiotics, 2022, doi:10.3390/antibiotics11101300_

Round 1
Reviewer 1 Report
In this study, the authors describe the elution capacity of dalbavancin in bone cement, presenting this drug as a possible alternative to vancomycin for the local antibiotic delivery in the treatment of PJIs.
Overall, the paper is well written, the description of methods and the presentation of results are clear and the topic is of interest for clinicians involved in the treatment of PJIs. Nevertheless, in my opinion more caution is needed in presenting dalbavancin as a potential option for antibiotic-loaded spacers.
Below are my considerations:
- Despite the use of antibiotic-loaded cement is widely accepted in the management of PJIs, there are some concerns about this practice, but only one of them (impairment of mechanical properties of cement by addition of antibiotic) is cited in the discussion section. To be thorough, also the other potential complications of the use of antibiotic-loaded cement spacers (i.e. risks of systemic toxicity and promotion of antibiotic resistance, especially in case of prolonged spacer retention, which may lead to sub-inhibitory concentrations of the antibiotic in the site of infection) should be mentioned.
- Due to the very long half-life of dalbavancin and the fact that in clinical practice cement spacers stay in place for more than 14 days, it would be very interesting to know what happen to dalbavancin concentration even after 14 days. If authors have this information it would be useful to include it.
- Lines 102-103: Reference 34 (Sanz-Ruiz P, Matas-Diez JA, Villanueva-Martínez M, et al. Is Dual Antibiotic-Loaded Bone Cement More Effective 265 and Cost-Efficient Than a Single Antibiotic-Loaded Bone Cement to Reduce the Risk of Prosthetic Joint Infection 266 in Aseptic Revision Knee Arthroplasty? J Arthroplasty. 2020) is not very pertinent, as it refers to the use of antibiotic-loaded bone cement to prevent PJI while the topic of the paper is its application in the treatment of PJIs by two-stage prosthesis exchange.
- Lines 136-138: authors say that further studies are needed “to test its efficacy against Staphylococcal biofilms using in vitro models 137 of bone cement elution”. They should indicate that before considering the use of dalbavancin for antibiotic-loaded bone spacers in clinical practice also its impact on mechanical properties of cement and potential toxicity should be further investigated.
Minor:
- The first paragraph of the introduction is not very clear, an English revision should be done
- Line 97: I would replace the word “demonstrating” with “describing”
- Figure 2 may be considered redundant, as the description of the experiment done in the text is clear enough. I would rather include the image now presented as supplementary material
Author Response
In this study, the authors describe the elution capacity of dalbavancin in bone cement, presenting this drug as a possible alternative to vancomycin for the local antibiotic delivery in the treatment of PJIs.
Overall, the paper is well written, the description of methods and the presentation of results are clear and the topic is of interest for clinicians involved in the treatment of PJIs. Nevertheless, in my opinion more caution is needed in presenting dalbavancin as a potential option for antibiotic-loaded spacers.
Below are my considerations:
- Despite the use of antibiotic-loaded cement is widely accepted in the management of PJIs, there are some concerns about this practice, but only one of them (impairment of mechanical properties of cement by addition of antibiotic) is cited in the discussion section. To be thorough, also the other potential complications of the use of antibiotic-loaded cement spacers (i.e. risks of systemic toxicity and promotion of antibiotic resistance, especially in case of prolonged spacer retention, which may lead to sub-inhibitory concentrations of the antibiotic in the site of infection) should be mentioned.
We agree with the reviewer and we have included some comments regarding these aspects in the discussion.
- Due to the very long half-life of dalbavancin and the fact that in clinical practice cement spacers stay in place for more than 14 days, it would be very interesting to know what happen to dalbavancin concentration even after 14 days. If authors have this information it would be useful to include it.
We are starting new experiments of dalbavancin activity against Staphylococcal biofilms for a 3-month period. At the end of our experiments, we will prepare a new publication regarding this aspect.
- Lines 102-103: Reference 34 (Sanz-Ruiz P, Matas-Diez JA, Villanueva-Martínez M, et al. Is Dual Antibiotic-Loaded Bone Cement More Effective 265 and Cost-Efficient Than a Single Antibiotic-Loaded Bone Cement to Reduce the Risk of Prosthetic Joint Infection 266 in Aseptic Revision Knee Arthroplasty? J Arthroplasty. 2020) is not very pertinent, as it refers to the use of antibiotic-loaded bone cement to prevent PJI while the topic of the paper is its application in the treatment of PJIs by two-stage prosthesis exchange.
We have modified the sentence by explaining that dual antibiotic high-loaded bone cements are demonstrated to be more effective and cost efficient in the prevention of PJI.
- Lines 136-138: authors say that further studies are needed “to test its efficacy against Staphylococcal biofilms using in vitro models 137 of bone cement elution”. They should indicate that before considering the use of dalbavancin for antibiotic-loaded bone spacers in clinical practice also its impact on mechanical properties of cement and potential toxicity should be further investigated.
The reviewer is correct, we have included this new aspect to be considered for further investigation.
Minor:
- The first paragraph of the introduction is not very clear, an English revision should be done.
English had been reviewed by an expert native speaker. However, we have change the paragraph in order to be clearer.
- Line 97: I would replace the word “demonstrating” with “describing”
We have done so as suggested.
- Figure 2 may be considered redundant, as the description of the experiment done in the text is clear enough. I would rather include the image now presented as supplementary material
We have converted figure 2 to supplementary material S2 as suggested.
Reviewer 2 Report
The authors presented an interesting subject, but all sections should be considered to be revised:
- A major revision of the English used is required throughout the manuscript.
- It looks more like a short communication than an article, because it is very short. Perhaps in the title the authors can add “in vitro study”, if they consider suitable and improving it.
- The abstract creates confusion as it is formulated. The second part of the sentence in line 18 looks like an aim or a conclusion, not here. In line 20, put the manufacturer of Palacos®R. Also, in line 23 specify whether the shaking occurred, during the night?, or correct the expression of the sentences in lines 21-24. On line 26, "than for" is not the correct English. Please respect the full names rule before using acronyms (PBS, HPLC and MIC).
- In line 33, the word pmma is unclear; it should be correct.
- The introduction section should be revised, especially the language used, must be scientific. Two subsequent sentences cannot begin with the same preposition "Among". In line 56 it is better to choose another suitable word instead of “vehicles”, for example “substances”, or other synonyms. Lines 70-73 seem more like a discussion than an introduction; please revise! Enter the full name of PMMA and then use this acronym in the following sentences.
- The results section must be organized into subsections according to the materials and methods section, explaining in detail and clearly the results obtained and not in general. This is a new in vitro study and its importance should be emphasized. Put the full names of SD and MIC.
- The discussion must be reformulated from the initial, because it creates confusion; for example, the sentences in lines 110-115, talk for themselves as authors???, so not clear, but in meantime they insert references. They need to be clearer if they want to discuss with other studies. It is better to add the limitations at the end of the discussion.
- Materials and methods should be organized into subsections and better explain the procedures followed for this in vitro study. A few lines and a scheme are not enough. On line 146, add the full name of PBS. It is correct to keep the same terminology, in the abstract is used "shaking", in line 147 is "agitation", in addition to specifying whether it was overnight or not or to correct the formulation of the following sentences. Add the full name of HPLC. For Nucleosil insert ® and also add the manufacturer. On lines 161-162, add the full name of LC-ESI-QQQ-MS and the city and state of Shimadzu. Also put the full name of MRM. The referee did not observe any "Statistical Analysis" section; not needed for the preparation of the table and the figure?
- The single sentence of conclusions is not sufficient to demonstrate the importance of this new study; it seems more like part of the discussion. It is better to cancel it and put the right conclusion.
- Suggestion: the referee does not consider Figure S1 suitable; it is very elementary and more for scholar than for publication. Perhaps it can be converted into a diagram/scheme, where the procedure for preparing the antibiotic-loaded bone cements is demonstrated.
Author Response
The authors presented an interesting subject, but all sections should be considered to be revised:
- A major revision of the English used is required throughout the manuscript.
The English had bed reviewed by Thomas O’Boyle, an expert native speaker of scientific reports, and we consider that English used in the manuscript is appropriate.
- It looks more like a short communication than an article, because it is very short. Perhaps in the title the authors can add “in vitro study”, if they consider suitable and improving it.
We have changed the study title to: “Determination of the elution capacity of dalbavancin in bone cements: New alternative for the treatment of biofilm-related peri-prosthetic joint infections based on an in vitro study”.
- The abstract creates confusion as it is formulated. The second part of the sentence in line 18 looks like an aim or a conclusion, not here. In line 20, put the manufacturer of Palacos®R. Also, in line 23 specify whether the shaking occurred, during the night? or correct the expression of the sentences in lines 21-24. On line 26, "than for" is not the correct English. Please respect the full names rule before using acronyms (PBS, HPLC and MIC).
We have now made the suggested corrections in the abstract. Regarding the question of when the shaking occurred, we consider that it has no relevance. Indeed, as we described later, aliquots were obtained at different times during incubation with shaking. However, we changed “with” to “under” to better explain the experiment.
- In line 33, the word pmma is unclear; it should be correct.
We have provided the full name.
- The introduction section should be revised, especially the language used, must be scientific. Two subsequent sentences cannot begin with the same preposition "Among". In line 56 it is better to choose another suitable word instead of “vehicles”, for example “substances”, or other synonyms. Lines 70-73 seem more like a discussion than an introduction; please revise! Enter the full name of PMMA and then use this acronym in the following sentences.
We have made the requested changes. However, we consider that the sentences 70-73: “Due to its pharmacokinetic characteristics, and in particular its long half-life (372 hours), dalbavancin can be administered, at least, once a week, following a regimen of one 1000 mg and one 500 mg dose spaced 7 days apart”, are worth of importance to be in the introduction as it describes the advantage of the antibiotic under research.
- The results section must be organized into subsections according to the materials and methods section, explaining in detail and clearly the results obtained and not in general. This is a new in vitro study and its importance should be emphasized. Put the full names of SD and MIC.
Despite now we have included subsection in material and methods as suggested below, no more detailed can be provided in the results section, as we only tested vancomycin and dalbavancin concentrations along a 14-day period time and this is what we have described in the results section.
- The discussion must be reformulated from the initial, because it creates confusion; for example, the sentences in lines 110-115, talk for themselves as authors???, so not clear, but in meantime they insert references. They need to be clearer if they want to discuss with other studies. It is better to add the limitations at the end of the discussion.
The sentences in lines 110-115 are the limitations previously described by other authors of the antibiotic-loaded cements, that is why references were inserted.
- Materials and methods should be organized into subsections and better explain the procedures followed for this in vitro study. A few lines and a scheme are not enough. On line 146, add the full name of PBS. It is correct to keep the same terminology, in the abstract is used "shaking", in line 147 is "agitation", in addition to specifying whether it was overnight or not or to correct the formulation of the following sentences. Add the full name of HPLC. For Nucleosil insert ® and also add the manufacturer. On lines 161-162, add the full name of LC-ESI-QQQ-MS and the city and state of Shimadzu. Also put the full name of MRM. The referee did not observe any "Statistical Analysis" section; not needed for the preparation of the table and the figure?
We have now separate methods section into subsections. Procedures followed in the in vitro study can no longer be better explained. We have added the full name of PBS. We have changed “agitation” to “shaking” and we have better clarified when it was performed as follows: “since the preparation of the cement discs were done until 14 days”, so it has no relevance whether shaking was overnight or not, as shaking was performed during 14 consecutive days and nights. We have added the full name of HPLC, the manufacturer of Nucleosil insert, the full name of LC-ESI-QQQ-MS, the city and state of Shimadzu, and the full name of MRM. We apologize for the mistake, as we forgot to include the statistical analysis section. We have now added in the material and methods section.
- The single sentence of conclusions is not sufficient to demonstrate the importance of this new study; it seems more like part of the discussion. It is better to cancel it and put the right conclusion.
We have now try to make a better conclusion statement.
- Suggestion: the referee does not consider Figure S1 suitable; it is very elementary and more for scholar than for publication. Perhaps it can be converted into a diagram/scheme, where the procedure for preparing the antibiotic-loaded bone cements is demonstrated.
We have converted figure S1 to a diagram as suggested.
Reviewer 3 Report
This is the good-written manuscript on a topic that is relevant. I have no critical remarks. I have several optional suggestions on how to improve the manuscript:
1. The title of the manuscript refers to biofilm-forming microorganisms, and the manuscript itself mentions gram-positive microorganisms. But more often biofilms are formed by gram-negative microorganisms. Please add your vision of prevention gram-negative microorganisms infections, or detail why you believe that gram-positive ones are more likely to form biofilms in case of join prothesis infections.
Author Response
This is the good-written manuscript on a topic that is relevant. I have no critical remarks. I have several optional suggestions on how to improve the manuscript:
- The title of the manuscript refers to biofilm-forming microorganisms, and the manuscript itself mentions gram-positive microorganisms. But more often biofilms are formed by gram-negative microorganisms. Please add your vision of prevention gram-negative microorganisms’ infections, or detail why you believe that gram-positive ones are more likely to form biofilms in case of join prosthesis infections.
We thank the reviewer the positive appreciation of the manuscript.
As title is a more general description, we preferred to maintain the term “biofilm” rather than “gram-positive biofilm”, since we have also included: “based on an in vitro study” in the title and we want not to extend it too much. It is well known that the main causative agents of PJI are gram-positive, so that is what we did not mentioned gram-negative. Moreover, as dalbavancin is al glycolipopetide only active against gram-positive bacteria, it is no relevant to mention PJI caused by gram-negative bacilli. We have included a brief description of the PJI ethology in the introduction.
Reviewer 4 Report
Please include Data is presented as number (proportion) in the footnote of tables.
The results are quite scarce. This translates to most of your discussion actually being appropriate for introduction and only last paragraf of discussion being actual discussion of your work. Only thing that can actually be discussed is similar PK. You may consider adding another sentence to the conclusion commenting on the similarities between the two antibiotics you researched in your experiment.
Furthermore, limitations are missing.
Supplementry file is silly and unnecessary. The paper seems unfinished, it would be complete should you have investigated "its efficacy against Staphylococcal biofilms using in vitro models of bone cement elution"
Author Response
Please include Data is presented as number (proportion) in the footnote of tables.
Data in tables are presented as mean ±SD (described in the column), not as number (proportions).
The results are quite scarce. This translates to most of your discussion actually being appropriate for introduction and only last paragraph of discussion being actual discussion of your work. Only thing that can actually be discussed is similar PK. You may consider adding another sentence to the conclusion commenting on the similarities between the two antibiotics you researched in your experiment.
Ours is a proof-of-concept study in which we only want to provide the novel information of the elution capacity of dalbavancin, since no previous studies have previously reported it. We consider that the content of the discussion about the possible benefits of dalbavancin used as loaded-bone cement, based on our results, is appropriate. We have already discussed the similar PK. However, the important thing to be highlighted was not that vancomycin and dalbavancin have similar PK, but the possible advantage of dalbavancin, based on its higher MIC90. However, we have included the following sentence regarding the similarities between both antibiotics in the conclusion: “as similar elution cumulative concentration was observed to that of vancomycin”.
Furthermore, limitations are missing.
We have now included a limitation section.
Supplementary file is silly and unnecessary. The paper seems unfinished; it would be complete should you have investigated "its efficacy against Staphylococcal biofilms using in vitro models of bone cement elution"
We consider that our study work needed to be divided into two different experiments: the first one as a proof of concept study assessing the potential elution capacity of dalbavancin when loaded in bone cements, and the second one as a more exhaustive study assessing the anti-biofilm efficacy of dalbavancin loaded in bone cement during a long period of time.
Round 2
Reviewer 2 Report
The authors improved their manuscript in this round!
Please italicize "in vitro". Regarding “shaking”, the referee has experience in laboratory procedures and suggests improving the expression on incubation time in line 24 (abstract) and lines 159-160; for example, by deleting the expression “Aliquots were obtained at the following incubation times:” and continuing the sentence by adding “for 1, 2, 4, 8, 24, 48, 168 and 336 h”. This makes more sense to combine the two steps. In line 108, correct "prevention".
In line 159, it is better to find another verb to express the preparation of the discs, it is not scientifically "were done". Please improve the quality of the Figures S1 and S2; also express in hours 7 and 14 days in S2.
Author Response
The authors improved their manuscript in this round!
Please italicize "in vitro". Regarding “shaking”, the referee has experience in laboratory procedures and suggests improving the expression on incubation time in line 24 (abstract) and lines 159-160; for example, by deleting the expression “Aliquots were obtained at the following incubation times:” and continuing the sentence by adding “for 1, 2, 4, 8, 24, 48, 168 and 336 h”. This makes more sense to combine the two steps. In line 108, correct "prevention".
In line 159, it is better to find another verb to express the preparation of the discs, it is not scientifically "were done". Please improve the quality of the Figures S1 and S2; also express in hours 7 and 14 days in S2.
Thank you for the suggestions, we have corrected all issues in the new version.
Reviewer 3 Report
Please, write `in vitro` in italic
Author Response
Reviewer 3.2
Please, write `in vitro` in italic.
Thank you, we have done so as suggested.